# Experimental Inoculation of *Aggregatibacter actinomycetemcomitans* and *Streptococcus gordonii* and Its Impact on Alveolar Bone Loss and Oral and Gut Microbiomes

**DOI:** 10.3390/ijms25158090

**Published:** 2024-07-25

**Authors:** Catarina Medeiros Rocha, Dione Kawamoto, Fernando Henrique Martins, Manuela Rocha Bueno, Karin H. Ishikawa, Ellen Sayuri Ando-Suguimoto, Aline Ramos Carlucci, Leticia Sandoli Arroteia, Renato V. Casarin, Luciana Saraiva, Maria Regina Lorenzetti Simionato, Marcia Pinto Alves Mayer

**Affiliations:** 1Department of Microbiology, Institute of Biomedical Science, University of São Paulo, São Paulo 05508-000, SP, Brazil; catarinamrocha@usp.br (C.M.R.); 77didi@gmail.com (D.K.); fernando.martins@alumni.usp.br (F.H.M.); manuela.bueno@slmandic.edu.br (M.R.B.); karinhitomi@hotmail.com (K.H.I.); esa.2406@gmail.com (E.S.A.-S.); alinecarlucci@usp.br (A.R.C.); mrsimion@usp.br (M.R.L.S.); 2Department of Stomatology, School of Dentistry, University of São Paulo, São Paulo 05508-000, SP, Brazil; 3Division of Periodontics, Faculdade São Leopoldo Mandic, São Leopoldo Mandic Research Institute, Campinas 13045-755, SP, Brazil; 4Department of Prosthesis and Periodontology, School of Dentistry, University of Campinas, Campinas 13083-875, SP, Brazil; leticia.sandoli@hotmail.com (L.S.A.); recasarin@unicamp.br (R.V.C.)

**Keywords:** periodontitis, *A. actinomycetemcomitans*, *Streptococcus gordonii*, alveolar bone loss

## Abstract

Oral bacteria are implicated not only in oral diseases but also in gut dysbiosis and inflammatory conditions throughout the body. The periodontal pathogen *Aggregatibacter actinomycetemcomitans* (*Aa*) often occurs in complex oral biofilms with *Streptococcus gordonii* (*Sg*), and this interaction might influence the pathogenic potential of this pathogen. This study aims to assess the impact of oral inoculation with *Aa*, *Sg*, and their association (*Aa*+*Sg*) on alveolar bone loss, oral microbiome, and their potential effects on intestinal health in a murine model. *Sg* and/or *Aa* were orally administered to C57Bl/6 mice, three times per week, for 4 weeks. *Aa* was also injected into the gingiva three times during the initial experimental week. After 30 days, alveolar bone loss, expression of genes related to inflammation and mucosal permeability in the intestine, serum LPS levels, and the composition of oral and intestinal microbiomes were determined. Alveolar bone resorption was detected in *Aa*, *Sg*, and *Aa+Sg* groups, although *Aa* bone levels did not differ from that of the SHAM-inoculated group. *Il-1β* expression was upregulated in the *Aa* group relative to the other infected groups, while *Il-6* expression was downregulated in infected groups. *A*a or *Sg* downregulated the expression of tight junction genes *Cldn 1*, *Cldn 2*, *Ocdn*, and *Zo-1* whereas infection with *Aa+Sg* led to their upregulation, except for *Cldn 1*. *Aa* was detected in the oral biofilm of the *Aa*+*Sg* group but not in the gut. Infections altered oral and gut microbiomes. The oral biofilm of the *Aa* group showed increased abundance of *Gammaproteobacteria*, *Enterobacterales*, and *Alloprevotella,* while *Sg* administration enhanced the abundance of *Alloprevotella* and *Rothia*. The gut microbiome of infected groups showed reduced abundance of *Erysipelotrichaceae*. Infection with *Aa* or *Sg* disrupts both oral and gut microbiomes, impacting oral and gut homeostasis. While the combination of *Aa* with *Sg* promotes *Aa* survival in the oral cavity, it mitigates the adverse effects of *Aa* in the gut, suggesting a beneficial role of *Sg* associations in gut health.

## 1. Introduction

Periodontitis involves multifactorial inflammatory processes that result in the destruction of tooth-support tissues, driven by dysbiotic microbiota present in supra- and subgingival biofilms [1]. Specific periodontopathogenic bacteria play a key role in the composition of dental biofilm, leading to heightened proportions of inflammatory organisms and increased virulence potential [2].

*Aggregatibacter actinomycetemcomitans* (*Aa*) is a Gram-negative, facultative anaerobic rod primarily associated with rapidly progressing periodontitis in young individuals, previously known as aggressive periodontitis (AP) [3]. This organism is able to subvert host response due to the production of a leukotoxin and a cytolethal distending toxin (CDT) [4]. Leukotoxin deactivates neutrophils, monocytes, and endothelial cells in humans and certain non-human primates but does not affect experimental rodent models [5,6]. *Aa*CDT disrupts the cell cycle of epithelial cells and fibroblasts, induces apoptosis in T lymphocytes, and impairs macrophages’ phagocytic activity [7,8,9]. Additionally *Aa*LPS triggers the release of inflammatory cytokines such as IL-1, IL-6, and TNFα [10,11], while the outer membrane protein OMP29 facilitates invasion of non-phagocytic cells [12] and exerts inhibitory effects on the expression of several inflammatory mediators [13].

Periodontitis, including the rapidly progressing form associated with *Aa*, is marked not only by oral dysbiosis but also by changes in the fecal microbiome [14,15,16]. Experimental evidence suggests that gut dysbiosis may arise following the ingestion of oral bacteria, resulting in intestinal barrier disruption and systemic inflammation [17,18,19,20,21].

During homeostasis, the integrity of the gut mucosa relies on the expression of transmembrane molecular complexes like tight junction (TJ) proteins and mucus production [22,23]. Alterations in the gut microbiota have been linked to changes in intestinal permeability originated from inflammation-induced damage and shifts in the transcriptional profile of TJ proteins [24]. Prior studies have shown that oral pathogens like *P. gingivalis* and *Aa* can compromise the oral mucosa barrier [25,26,27] since *P. gingivalis* proteases degrade epithelial adhesion molecules [28] whereas AaCDT induces redistribution of E-cadherin and of proteins of the adherens junction complex within the epithelial cells and causes detachment of basal cells from the basement membrane [24].

The virulence and survival of bacteria from multispecies biofilms in the oral cavity are influenced by host factors and neighboring microorganisms. Periodontitis arises from the synergistic effect of the entire microbiota and a disease-associated bacterial consortium including *Aa* and *Streptococcus gordonii* (*Sg*) was reported in aggressive periodontitis [29]. The interaction between *Aa* and H_2_O_2_-producing streptococci results in altered virulence and increased survival of *Aa*, as demonstrated by in vitro biofilm studies and in a murine abscess model [30,31,32]. However, the potential of the *Aa*–streptococci consortium to induce experimental periodontitis and its role in in promoting microbial dysbiosis in the oral cavity and gut have not been evaluated.

Therefore, we aimed to assess the impact of the oral administration of *Aa*, *Sg*, and a consortium of *Aa* and *Sg* on alveolar bone resorption, expression of genes related to gut mucosa integrity, and oral and gut microbiomes, in a murine experimental model.

## 2. Results

### 2.1. Side Effects and Weight Gain Induced by Different Treatments

Four groups were evaluated, *Aa*-, *Sg*-, and *Aa+Sg*-infected groups along with the SHAM-inoculated group (negative control) (Figure 1). No changes in skin, fur, behavior, or mobility were observed in the animals throughout the 30-day experimental period. Initial weight was similar among groups. However, infection with *Aa* seemed to pose a detrimental effect on mice, since the Aa = infected groups (*Aa* and *Aa+Sg*) showed lower final weights compared to the *Sg* group (Figure 2A).

### 2.2. Alveolar Bone Loss Computed Microtomography (micro-CT)

Analysis of bone volume and porosity in the interproximal region between the first and second molars in the left hemimaxillae revealed that infection with *Sg* resulted in a lower percentage of bone volume and higher bone porosity compared to the other groups. However, alveolar bone loss values for the *Sg* group were similar to those observed for the *Aa+Sg* group. Infection with *Aa* alone did not significantly alter bone volume and porosity compared to the SHAM group, although the *Aa* group exhibited similar alveolar bone loss to the *Aa+Sg* group (Figure 2B,C).

### 2.3. LPS Levels in Serum

Serum LPS levels did not vary significantly among the groups, although the mean values of endotoxin units (EUs) were higher in the *Aa*-infected groups (*Aa+Sg* and *Aa*) compared to the *Sg* and SHAM groups.

### 2.4. Gene Expression Related to Gut Inflammation and Barrier Permeability

*Aa* infection resulted in higher expression of *Il-1β* compared to infection with *Sg* or *Aa+Sg*. Expression of *Il-6* was downregulated in the *Aa*- and *Sg*-infected groups but not in the *Aa+Sg* group compared to SHAM (Figure 3A,B). There were no differences in the transcription levels of *Il-10* and *Tnfα* among the groups.

Infection with *Aa* and/or *Sg* induced changes in the transcriptional profile of genes encoding TJ proteins claudin 1, claudin 2, and occludin and the anchoring protein zonulin in the intestinal barrier. Overall, infection with *Aa* or *Sg* downregulated the transcription of these genes, whereas infection with both organisms in the *Aa+Sg* group increased their mRNA levels, similar to the levels of the SHAM group, except for *Cldn1*, which achieved lower mRNA levels in *Aa* and *Aa+Sg* groups (Figure 3C–F). Transcription of mucin encoding genes was anot altered by the treatments.

### 2.5. Oral and Gut Bacterial Community Structure

The oral biofilm and gut samples underwent *16S rRNA* sequencing analysis. DNA was successfully extracted from oral and gut samples of 24 animals (6 per group). However, two gut samples from the SHAM group failed to yield amplicons for analysis, resulting in microbiome data of only four SHAM samples. A total of 7,207,606 sequences were obtained (ranging from 97,250 to 246,890). After filtering and removing chimeras, 2,719,335 sequences remained for oral samples (ranging from 87,139 to 150,712) and 390,140 for gut samples (ranging from 4854 to 32,775). Rarefaction curves based on the number of operational taxonomic units (OTUs) indicated that all samples reached a plateau when oral biofilm samples were normalized to 10,000 amplicon sequence variants (ASVs), and gut samples were normalized to 4000 sequences.

Faith’s phylogenetic diversity index analysis revealed that inoculation with *Sg* increased the diversity of the oral microbiome compared to *Aa* [33], although they did not differ from SHAM. In contrast, the concomitant administration of *Aa*+*Sg* reversed this effect (Figure 4A). Pielou (evenness) and Shannon (richness) diversity index analyses for the oral microbiome did not result in significant differences among the groups.

Analysis of the gut microbiome revealed differences in evenness index (Pielou) among the three infected groups [34] but not from the SHAM-inoculated group (Figure 4B). Inoculation with *Aa* or *Sg* resulted in reduced diversity compared to inoculation with both organisms in the *Aa+Sg* group. There were no differences in Faith and Shannon indexes among the groups for the gut microbiome.

Beta diversity analysis based on weighted and unweighted UniFrac distances revealed that the treatments affected both the oral and gut microbiomes. Infection with *Aa* or *Sg* altered the structure of the oral microbiome, while inoculation with *Aa+Sg* did not result in significant changes compared to SHAM (Figure 5A(I,II)). Analysis of gut samples showed clear segregation among the infected groups and SHAM. However, the gut microbiomes of *Aa*, *Sg*, and *Aa+Sg* did not differ from each other, except for differences between *Aa+Sg* and *Sg* based on unweighted UniFrac distances (Figure 5B(I,II)).

There were no differences in the abundance of different phyla in the oral or gut microbiomes among the groups (Figure 6 and Figure 7, respectively). Firmicutes was the most abundant phylum in the oral microbiome, followed by Pseudomonadota. ANCOM revealed that infection with *Aa* led to an increased abundance of Gammaproteobacteria, Enterobacterales, and the genus *Alloprevotella*, while infection with *Sg* resulted in increased abundance of the genera *Alloprevotella* and *Rothia* in the oral cavity. Infection with both organisms did not lead to significant changes in the oral microbiome (Figure 6).

Firmicutes was the most abundant phylum in the gut microbiome (Figure 7). Inoculation of *Aa*, *Sg*, or both microorganisms in the oral cavity resulted in decreased abundance of the order Erysipelotrichales and the family *Erysipelotrichaceae* compared to SHAM in the gut.

eHOMD database analysis revealed that *Aa* was detected in three out of six oral biofilm samples from the *Aa+Sg* group, but not in the *Aa* group, at an abundance lower than 0.001. *Sg* was not detected in the oral cavity of any mice. None of the inoculated species was detected in the gut.

## 3. Discussion

Periodontitis is associated with a dysbiotic microbiome. The microbial community within any host is influenced not only by the neighboring microorganisms but also by host defense, environmental, genetic, and epigenetic factors [35]. However, not all these factors can be replicated in a murine model. Additionally, animal models for studying periodontitis are limited by the specificity of certain organisms and their virulence factors to humans.

Therefore, our findings should be interpreted within the constraints of the animal model utilized. Colonization represents a crucial phase in infectious diseases, yet *Aa* oral colonization in mice is hindered due to the inability of *Aa* adhesins Aae and OMP100 to bind to murine epithelial cells [36,37]. In addition, *Aa* leukotoxin targets only human and certain non-human primate monocytes, macrophages, and neutrophils [38]. It is also important to notice that the resident microbiota of rodents can be easily influenced by mouse lineage but also by various factors, including breeding sources and conditions [39], altering the results of assays performed at different animal facilities.

The mice infected with *Aa* present a lower weight at the end of the experimental period when compared with those infected with *Sg* or SHAM-inoculated mice. This observation differs from studies in humans where detection of *Aa* in the oral microbiome was associated with obesity [40]. Experimental administration of *Aa* in mice fed a high-fat diet also resulted in increased weight gain compared to non-infected mice, whereas infection had no effect in weight gain in animals under a normal chow diet [21]. However, any comparison between our data and the latter study is prevented by different tested protocols, especially by the use of norfloxacin, an antimicrobial that modulates the gut microbiota and leads to improved fasting glycemia and oral glucose tolerance [41].

Despite these limitations, our data revealed that the interaction with *Sg* increased the ability of *Aa* to colonize the oral biofilm of mice, since half of the animals of the *Aa+Sg* group harbored *Aa* in their oral cavity after 30 days of the experimental period, whereas this species was not detected in any mouse of the *Aa* group. *Sg* increases *Aa* fitness by providing lactate for *Aa* and producing H_2_O_2_, enhancing oxygen bioavailability which alters the metabolism from fermentation to respiration, changes the transcription profile of the pathogen [6,32], and promotes its virulence in an abscess model [31,42]. Moreover, *Aa* dispersal in biofilms is dependent on dispersin B, whose expression is upregulated in co-infection with *Sg*, as shown by in vitro data [31].

Our data evidenced the relevance of the interaction with *Sg* to increase *Aa* survival in the oral biofilm of rodents. A single study in rats reported that the oral inoculation of *Aa* resulted in successful colonization of the oral biofilm and induction of periodontitis in wild type rodents [43]. Therefore, most studies on experimental alveolar bone loss in rodents induced by *Aa* inoculated the living organism or its products in the gingival tissue [44,45,46,47,48], resulting in *Aa* persistent colonization of the gingival tissues but not of the oral biofilm [46]. This association of *Aa* with lactate producers in the oral cavity is not only relevant for the murine model but has already been reported in experimental infection of *Aa* in primates. However, *Aa* recovery was low even in these primates after 4 weeks [49].

We have also shown that the three infected groups exhibited alveolar bone resorption, notwithstanding the similar bone volume and porosity observed in the *Aa* group compared to SHAM and to *Aa+Sg*. Therefore, although the lactate/peroxide producer streptococci provided increased survival of *Aa* in the oral biofilm, this association did not enhance the destruction of the periodontal tissues. It is worth mentioning that the bottom of deep periodontal pockets in humans represents a highly anaerobic environment where peroxide production is limited by the low concentration of oxygen, and *Aa* metabolism is anaerobic [32]. Furthermore, microbiome data indicated that the increased abundance of *Aa* in deep periodontal pockets of periodontitis patients is accompanied by reduced levels of streptococci and other lactate producers [16].

The oral administration of *Sg* also resulted in alveolar bone loss, either independently or in conjunction with *Aa* in the murine model. *Sg* is considered an accessory pathogen in the oral cavity due to its interactions with pathogens [50], despite its relatively low prevalence compared to other H_2_O_2_ producers streptococci in the oral cavity [51]. This species can activate osteoclasts [52] and induce an inflammatory response in experimental apical periodontitis [53], stimulating the production of chemokines such as CXCL8 by periodontal ligament fibroblasts [54] as well as the maturation of dendritic cells [55]. Thus, under the limitations of the murine model, it seems that *Sg* may be important for *Aa* survival in the supragingival biofilm, but the co-infection may not be relevant for *Aa*-induced bone resorption.

Although *Sg* facilitated the recovery of *Aa* in the oral cavity, the impact of oral administration of *Aa* in the gut was more pronounced compared to that promoted by the *Aa+Sg* consortium. Limited knowledge exists regarding the role of *Aa* and/or *Sg* as inducers of intestinal dysbiosis and alteration of the intestinal barrier. *Aa* infection induced the expression of *Il-1β* in the gut, while *Sg* or the *Aa+Sg* consortium did not. *Il-1β* is commonly associated with heightened intestinal permeability [56]. Conversely, *Sg* mitigated the *Aa*-induced inflammatory response in the gut, evidenced by the low transcriptional levels of *Il-1β* in the *Aa+Sg* group. Previous data have indicated that H_2_O_2_ produced by streptococci activates the Nrf2 pathway in macrophages, inhibiting the NFκB transcription factor and reducing the production of proinflammatory cytokines [57]. Other factors may also have contributed to this disparity. *Aa* mono-infection, but not the dual infection with *Aa* and *Sg*, resulted in the expression of virulence factors such as the cytolethal distending toxin (CDT) [32], which is known to activate the inflammasome leading to IL-1β release [58]. Furthermore, H_2_O_2_ acts as a signal to trigger the expression of catalase and OMP100 (ApiA) by *Aa*, which are involved in defense against the host’s innate immune system [59]. In contrast, the IL-1β pathway may hinder *Aa* survival in the gut [60].

We have also demonstrated downregulation of *Il-6* in the *Aa*, *Sg*, and *Aa+Sg* groups compared to the SHAM group. IL-6 plays a role in leukocyte recruitment during the acute inflammatory response by selectively regulating inflammatory chemokines and apoptotic events [17,61,62]. Therefore, it appears that infection with *Aa* and *Sg* contributed to the control of the inflammatory process, likely facilitated by the release of H_2_O_2_ by the accessory pathogen *Sg* [50], although other factors, such as additional regulatory proteins such as *Aa* outer membrane protein OMP29, may also play a role [13].

It is well recognized that dysregulation of the mucosal barrier facilitates the translocation of bacterial components or living cells [56] from the intestinal lumen through the epithelium, triggering immune activation and leading to systemic inflammation [63,64]. Alterations in the expression of TJ proteins can compromise the integrity of the barrier, promoting increased intestinal permeability, thus resulting in a condition often referred to as “leaky gut” [63,65,66].

Oral infection with *Aa* and/or *Sg* resulted in altered transcription of several genes involved in gut barrier integrity [67]. Generally, *Aa* or *Sg* downregulated the expression of *Cldn 1* and *Cldn 2*, *Ocdn*, and *Zo-1*, whereas the *Aa+Sg* consortium upregulated the expression of *Cldn 2*, *Ocdn*, and *Zo-1*. The elevated levels of *Il-1β* transcripts and low levels of *Il-6* transcripts in the gut of *Aa*-infected mice may partially account for this regulation induced by the pathogen [68]. Microbial insults and the resulting inflammatory response are known to contribute to the regulation of TJ expression and localization. Expression of Claudin 1 diminishes upon chronic exposure to LPS challenge [69], whilst Claudin 2 expression is associated with protection of microbial-induced colitis [38]. IL-6 upregulates expression of Claudin *2*, while TNF and IL-1β indirectly trigger occludin removal from the tight junction and increase leaky pathway permeability. On the other hand, IL-10 and TGFβ promote epithelial proliferation and restitution, leading to barrier restoration [70]. Thus, the attenuation of *Il-1β* and *Il-6* regulation observed when *Aa* was combined with *Sg* may contribute to the enhanced expression of TJs in the gut barrier. Previous data reported that *Sg* exerts a beneficial effect on the epithelial barrier function, particularly in gingival cells, by inhibiting cytokine secretion and enhancing the expression of key tight junction components, thus reinforcing the oral barrier against microbial invasion [71]. However, it should be noticed that the changes in TJ transcription profiles induced by *Aa* and/or *Sg* were discreet, since we could not detect endotoxemia in any of the studied groups, an indicator of altered gut permeability [72].

Oral inoculation of *Aa, Sg*, or *Aa+Sg* led to alterations in both oral and gut microbiomes. The oral microbiota plays a critical role in maintaining oral health and can influence intestinal dysbiosis. In periodontitis, dysbiosis has been observed not only in the oral cavity but also in the intestine [14,15,16]. The mechanism behind this phenomenon involves the ingestion of oral bacteria, which may persist in the gut for at least 24 h [73], potentially inducing dysbiosis [74].

Our data indicate that oral administration of *Sg* increased the diversity of the oral microbiome (Faith phylogenetic diversity index), whereas *Aa* or the consortium *Aa+Sg* reduced the diversity when compared to mono-infection with *Sg*. Phylogenetic diversity accounts for the phylogenetic relatedness of the community members [33] and reduced diversity induced by *Aa* means that this organism puts pressure on the oral microbiome in order to select a subset of organisms. These findings are consistent with studies showing a decrease in the microbial diversity of the oral cavity in periodontitis patients [75,76], indicating a role for *Aa* in oral dysbiosis.

The population structure of the oral microbiome differed between the *Aa*-infected group and the *Sg*-infected group compared with the SHAM group, with all infected groups differing from each other. The oral microbiome of *Aa*-infected mice exhibited increased abundance of Pseudomonadota, class Gammaproteobacteria, order Enterobacterales, and of the genus *Alloprevotella*. *Sg* administration also increased the abundance of *Alloprevotella* but additionally led to higher levels of *Rothia*. The implications of these microbial shifts are not fully understood, as members of Enterobacterales and the family *Prevotellaceae* comprise species associated with both disease and health [77,78]. However, Enterobacterales has previously been associated with Crohn’s disease [79,80,81,82], whereas *Rothia* is typically associated with health [83].

Animals infected with *Aa*, *Sg*, or both organisms exhibited a reduced abundance of *Erysipelotrichaceae* in the gut, reaching non-detectable levels. *Erysipelotrichaceae* is considered a beneficial organism to the gut (97) and previous data have shown that *Aa* oral inoculation resulted in lower abundance of *Turicibacter*, a genus of the family *Erysipelotrichaceae*, in feces [21]. Oral administration of streptococci in mice, such as *S. mitis* or *S. salivarius*, also reduced the levels of *Turicibacter* in stool samples [84]. The beneficial role of *Erysipelotrichaceae* in the gut appears to be due to the production of butyric acid, a fatty acid that provides energy to colonocytes and regulates the immune response [85]. Thus, our data suggest that infection with *Aa*, *Sg*, or both organisms results in depletion of beneficial organisms, in agreement with reduced transcription of genes associated with gut permeability.

Periodontitis has been linked to chronic inflammatory bowel disease (IBD) and colon cancer [86]. It is noteworthy that gut dysbiosis in patients with periodontitis, particularly the molar–incisor pattern associated with high abundance of *Aa*, was characterized by a greater abundance of sulfate-reducing bacteria (SRB), which are potential risk factors for colon cancer [16,87]. *Aa* can be found in low abundance in healthy individuals [88], but the oral microbiota in health harbors a higher proportion of H_2_O_2_ producers, such as *S. sanguinis* and *S. gordonii*, compared to disease [51]. Thus, the low transcriptional levels of *Il-1β* in the gut of mice of the *Aa+Sg* group compared to the *Aa* group suggest that these streptococci may limit the inflammatory potential of *Aa* in the gut. On the other hand, damage not only to the periodontium but also to the gut barrier appears to increase when the abundance of *Aa* rises due to host and bacterial factors, concurrent with decreased levels of lactate/H_2_O_2_ producers [89].

Taken altogether, our data indicate that *Aa* can disrupt both oral and gut microbiomes. The pathogen may disturb gut homeostasis by inducing the expression of inflammatory mediators such as IL-1β and by altering the expression profile of proteins related to intestinal barrier integrity. *Aa* co-exists in the oral cavity with other species such as *Sg*, which enhances its survival in the oral biofilm. However, co-infection of *Aa* with *Sg* seems to mitigate the effects of *Aa* on the gut.

## 4. Materials and Methods

### 4.1. Ethical Considerations

This study was conducted after obtaining approval from the Institutional Animal Care and Use Committee (ICB USP approval number: 4828281020), in accordance with the ethical principles of animal experimentation outlined by the Brazilian College of Animal Experimentation (COBEA).

### 4.2. Animals and Maintenance Conditions

C57BL/6 specific pathogen-free (SPF) male mice, aged 6–8 weeks, were obtained from the Animal Facility at the Faculty of Medicine and subsequently transferred to the Department of Microbiology and Parasitology at the Institute of Biomedical Sciences, University of São Paulo. They were housed in cages with a maximum of four mice per cage, fitted with microisolator filters, and placed on ventilated shelves to maintain pathogen-free conditions, with food (chow diet) provided ad libitum.

### 4.3. Sample Calculation and Group Allocation

Sample calculation was conducted for alveolar bone loss as the outcome, with a significance level (alpha) of 5%, power of 80%, and considering a 10% loss of animals during the experiment, based on our previous pilot data [90,91]. After a 7-day acclimation period in the facility, forty-eight (48) animals were randomly assigned to four groups, with twelve animals per group: SHAM, infected with *Aa*, infected with *Sg*, and infected with both species (*Aa+Sg*). The animals were then monitored for an experimental period of 30 days. Animals showing abnormalities in growth, weight, or physical characteristics at baseline were excluded from the study. Throughout the experiment, animals were monitored for changes in weight, mobility, coat condition, and presence of skin lesions.

### 4.4. Blinding

Each animal was assigned a temporary random number within its respective group. Cages were numbered based on their placement on the rack. A cage was randomly selected from each group from the available pool. Blinding procedures were maintained throughout the allocation process, evaluation of results, and data analysis. However, blinding was not possible during the experiment due to the same researcher being involved in both the preparation and inoculation of organisms. Additionally, the bacterial suspensions differed in color from the vehicles used.

### 4.5. Strains and Culture Conditions

*Aa* JP2 [38] and *Sg* DL1 [92] strains were utilized. All strains were maintained in 20% glycerol in brain–heart broth at −80 °C.

*Aa* was cultured in tryptone soy broth (Difco Laboratories, Detroit, MI, USA) supplemented with 0.5% yeast extract (Difco Laboratories) while *Sg* was cultured in BHI broth (Difco Laboratories), both incubated at 37 °C, in 5–10% CO_2_. For the assays, liquid medium was utilized to grow bacteria to mid-log phase. Subsequently, the cultures were centrifuged (355× *g* for 5 min), and the cell pellets were suspended in phosphate-buffered saline solution (1× PBS) to an optical density at 590nm of approximately 1, corresponding to 1 × 10^9^ CFU *Aa*/mL and to 1 × 10^12^ CFU *Sg*/mL.

The cell suspensions were centrifuged (6000× *g*) and cells suspended in a solution containing 10% skim milk powder (Molico, Nestle, Brazil), 5% sodium L-glutamate monohydrate (Sigma-Aldrich, Darmstadt, Germany), and 5% dithiothreitol (Sigma-Aldrich). The suspension was lyophilized using a freeze-dryer (Freezone Triad Freezer Dryers, Labconco, Flawil, Switzerland) at −4 °C under vacuum.

The lyophilized material was stored in a freezer at −80 °C. The viability of each batch of lyophilized bacteria was assessed by culturing them on agar plates under suitable conditions. Before each day of inoculation, the samples were reconstituted in BHI broth for one hour. Subsequently, the lyophilized bacteria were suspended in PBS at standardized concentrations.

### 4.6. Experimental Infections

Fifty-microliter aliquots containing 1 × 10^9^ CFU of *Aa* and/or 1 × 10^9^ CFU of *Sg* in PBS with 1.5% carboxymethylcellulose were inoculated into the oral cavity of mice using a gavage needle, three times a week, for four weeks, totalizing 12 inoculations [47]. On the first, third, and fifth days of the experimental period, the animals underwent intraperitoneal anesthesia with ketamine (100 mg/kg) (10% Dopalen, Vetbrands Laboratory, Rio de Janeiro, Brazil) and xylazine (10 mg/kg) (2% Ansedan, Vertbrands Laboratory), when they received an injection at the interproximal gingiva between the first and second molars in the left hemimaxilla of a volume of 10 µL containing 1 × 10^7^ CFU of *Aa* in PBS [48]. The vehicle (PBS with 1.5% carboxymethylcellulose) was orally administered to the SHAM-inoculated group. Both the SHAM and *Sg* groups were anesthetized and received gingival injections with PBS on the same days and with the same volumes as the other *Aa* infected groups. After 30 days, the mice were euthanized, and samples were collected. The study design is illustrated in Figure 1.

### 4.7. Euthanasia and Sample Collection

The animals were anesthetized intraperitoneally with ketamine (100 mg/kg) and xylazine (10 mg/kg). Blood was collected via cardiac puncture, and the mice were euthanized by exsanguination. Serum was obtained by centrifugation. The left hemimaxillae were transferred to formaldehyde solution and kept at room temperature for 24 h, then transferred to PBS and stored at 4 °C until analysis. The jejunal portions of the intestine were placed in RNAlater ^TM^ Stabilization Solution (Invitrogen by Thermo Fisher Scientific, Vilnius, Lithuania) and stored at −80 °C. Oral biofilms were collected using microbrushes, and jejunal content samples were obtained using a wooden spatula and stored in Tris-EDTA buffer, pH 7.4.

### 4.8. Computerized Microtomography (Micro CT)

Alveolar bone resorption was assessed using microtomography (SkyScan 1174 version 1.1, Kontich, Belgium) operating at 45 kV voltage, 550 μA current, with an 8.71 μm pixel size and a 0.2 mm aluminum filter. Scans were conducted on the left hemimaxillae, and a blinded evaluator identified a standardized area of 60 × 30 pixels (ROI) at the interproximal region between the first and second molars, originating from the cementoenamel junction of the second molar across 15 coronal sections. Bone volume and porosity percentages were determined through image analysis using CT Analyzer software version 1.15.4.0.

### 4.9. Serum Lipopolysaccharide Levels

Lipopolysaccharide (LPS) levels were quantified using the Pierce LAL Chromogenic Endotoxin Quantification Kit #88282 (Thermo Scientific™, Rockford, IL, USA). Solutions ranging from 0.1 to 1 endotoxin units (EUs) per milliliter of LPS Endotoxin Standard (*E. coli* 011: B4) were employed to determine the endotoxin concentration of each sample. Absorbance was measured at 405 nm using the Epoch™ 2 microplate spectrophotometer (BioTek, Winooski, VT, USA).

### 4.10. Gene Expression Linked to Gut Inflammation and Epithelial Barrier Permeability

Gut samples underwent RNA extraction using Trizol LS Reagent (Invitrogen Life Technologies, Carlsbad, CA, USA) with the assistance of a Mini-BeadBeater (BioSpec 3110BX Mini-BeadBeater-1 High Energy Cell Disrupter, Campinas, São Paulo, Brazil) for 20 s, repeated twice. The resulting RNA was treated with desoxyribonuclease (Ambion™ Dnase I, Invitrogen Life Technologies). Subsequently, cDNA was synthesized using the SuperScript TM Vilo TM Synthesis Kit for RT-PCR (Invitrogen Life Technologies). Quantitative PCR was conducted using the StepOne Plus System thermocycler (Applied Biosystems, Foster City, CA, USA).

Transcription levels of cytokine genes were determined using 100 ng of cDNA and TaqMan™ Gene Expression Assay (Invitrogen by Thermo Fisher Scientific, Vilnius, Lithuania) with commercial TaqMan primers and probes (Invitrogen Life Technologies) for *IL-1β* (Mm00434228_m1), *TNFα* (Mm00443258_m1), *IL-10* (Mm01288386_m1), *IL-6* (Mm00446190_m1), and *GAPDH* (Mm99999915_g1).

The expression of genes related to intestinal epithelial barrier permeability including *Claudin 1* (*Cldn1*), *Claudin 2* (*Cldn2*), *Occludin* (*Ocdn*), *Mucin 1* (*Muc1*), *Mucin 2* (*Muc2*), and *Zonulin* (*Zo-1*) was determined using primers synthesized by Thermo Fisher Scientific (Waltham, MA, USA) (Table 1). Each reaction consisted of 5 μL of SYBR Green (Thermo Fisher Scientific), 200 ng of cDNA, and 0.625 pM of each oligonucleotide in a final volume of 10 μL. Reactions were performed through forty cycles of 95 °C/15 sec, 65 °C/1 min, 81 °C/10 s, followed by two steps at 95 °C/15 s and 65 °C/1 min, and a final step at 0.5–95 °C/15 s.

The relative expression of target genes was calculated using the ΔΔCT method, utilizing *GAPDH* as an endogenous control [39] and expressed as fold changes relative to the control group (SHAM).

### 4.11. Oral and Gut Microbiomes

DNA from oral biofilm and gut samples was extracted using the Master PureTM DNA Purification Kit (Epicentre^®^ Illumina Company, Madison, WI, USA). The quality of purified DNA was assessed using the Qubit 2.0 fluorometer (Thermo-Fisher Scientific, Carlsbad, CA, USA). Unfortunately, DNA from the biofilms of two animals from each group was lost during manipulation, leading to the exclusion of gut samples from these animals, leading to microbiome analyses of six samples from each site per group.

A barcoded primer set, based on universal primers Bakt_341F (CCTACGGGNGGCWGCAG) and Bakt_805R (GACTACHVGGGTATCTAATCC) [98], was utilized to amplify the hypervariable V3–V4 region of the 16S rRNA gene. Subsequently, DNA samples were sequenced by ByMyCell (Ribeirão Preto, São Paulo, Brazil) using the Illumina MiSeq 2 × 250 platform. The sequence data were then submitted to the Sequence Read Archive (SRA) under BioProject identification numbers PRJNA995344 and PRJNA994574.

Data were analyzed using Qiime 2 2022.8 software [99]. Demultiplexed sequences and reads underwent filtering using Dada 2, with a quality threshold set at 25. Subsequently, trimmed sequences were clustered into amplicon sequence variants (ASVs), and taxonomy was assigned using the Silva138 database [100,101]. Alpha diversity indices, including Faith, Pielou, and Shannon, were assessed. Beta diversity analyses were conducted using weighted and unweighted UniFrac distance matrices and were visualized through principal coordinate analysis (PCA) [99]. Significant differences among groups were determined by PERMANOVA with 999 permutations. The discrepancies in abundances of ASVs among groups were analyzed using ANCOM [102] with a 75% empirical cutoff value for the w statistic [103]. To identify the inoculated species in the oral and gut microbiome, the Expanded Human Oral Microbiome Database (eHOMD) was utilized [104].

### 4.12. Data Analyses

The data were assessed for normality using the Kolmogorov–Smirnov statistical test with Lilliefors correction, and the homogeneity of variances was evaluated using the F test. For standard data analysis, the Kruskal–Wallis test followed by Dunn’s post hoc test was utilized. Statistical significance was set at *p* < 0.05. All analyses were conducted using GraphPad Prism^®^ version 6.0 software (GraphPad Software, La Jolla, CA, USA).

## Figures and Tables

**Figure 1 ijms-25-08090-f001:**
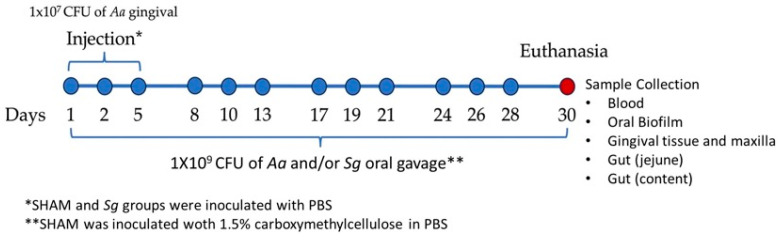
Study design: C57BL/6 mice were randomly assigned into four groups: the SHAM-inoculated group, the group infected with *Aa*, the group infected with *Sg*, and the group infected with both species, *Aa*+*Sg*. After a 30-day experimental period, the mice were euthanized, and samples were collected.

**Figure 2 ijms-25-08090-f002:**
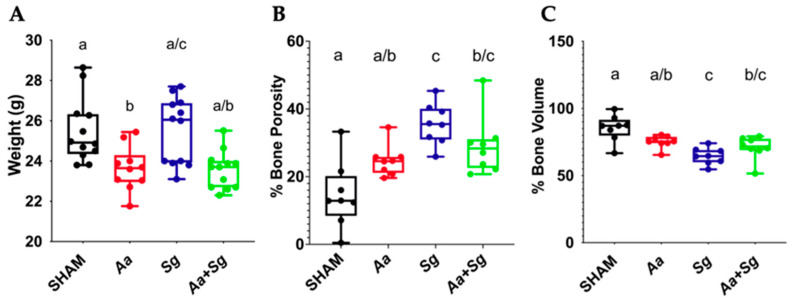
Analysis reveals alveolar bone resorption in C57BL/6 mice maxilla following various infection protocols (SHAM, *Aa*, *Sg*, *Aa*+*Sg*). Mice weight (**A**) and bone porosity (**B**) and bone volume (**C**) which were measured at the interproximal area between the first and second molars on the left hemimaxilla at the end of the experimental period. Distinct letters indicate statistically significant differences (*p* < 0.05) between groups (Kruskal–Wallis followed by post hoc Dunn test).

**Figure 3 ijms-25-08090-f003:**
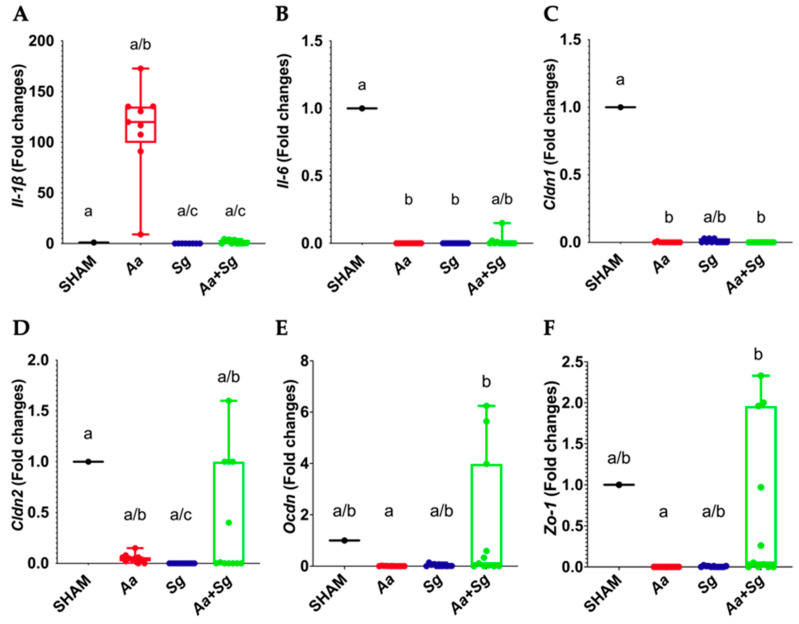
Relative transcription of genes involved in inflammation and intestinal permeability in the jejunum of C57BL/6 mice subjected to different infection protocols: SHAM (non-infected negative control), *Aa* (administered orally and via gingival injections), *Sg* (administered orally), and *Aa*+*Sg* (*Aa* and *Sg* administered orally and *Aa* via gingival injections), at the end of the experimental period. *Il-1β* (**A**), *Il-6* (**B**), *Cldn 1* (**C**), *Cldn 2* (**D**), *Ocdn* (**E**), and *Zo-1* (**F**). Relative gene transcription data are presented as fold changes compared to the negative control (SHAM). Distinct letters indicate statistically significant differences determined by Kruskal–Wallis followed by Dunn’s post hoc test (*p* < 0.05).

**Figure 4 ijms-25-08090-f004:**
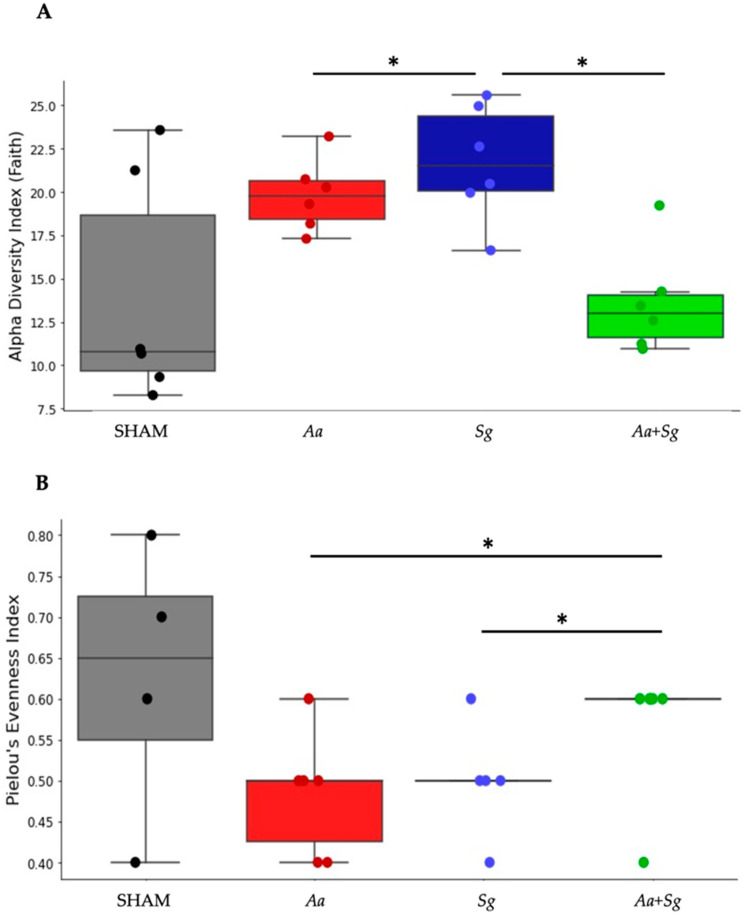
Faith diversity index of the oral microbiome (**A**) and Pielou diversity index of the gut microbiome (**B**) in C57BL/6 mice subjected to different infection protocols: SHAM (non-infected negative control), *Aa* (administered orally and via gingival injections), *Sg* (administered orally), and *Aa*+*Sg* (*Aa* and *Sg* administered orally and *Aa* via gingival injections), at the end of the 30-day experimental period. * Statistically significant differences (*p* < 0.05) between groups determined by Kruskal–Wallis followed by Dunn’s post hoc test.

**Figure 5 ijms-25-08090-f005:**
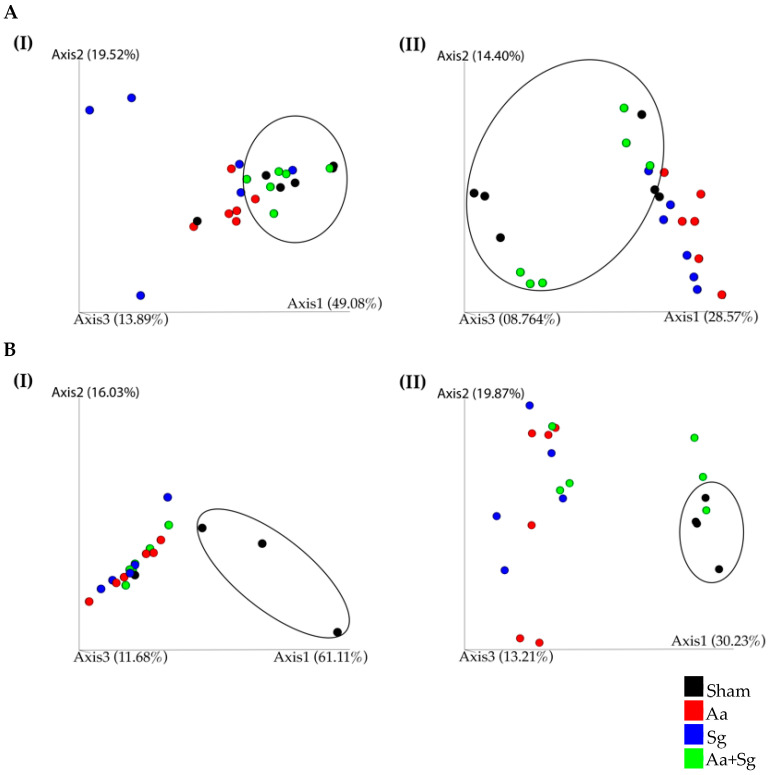
Principal coordinate analysis (PCoA) of oral (**A**) and gut (**B**) microbiomes based on weighted (**I**) and unweighted (**II**) UniFrac distance of C57BL/6 mice under various infection protocols: SHAM (negative control), *Aa* (oral and gingival *Aa* administration), *Sg* (oral *Sg* administration), and *Aa*+*Sg* (oral *Aa* and *Sg* administration and gingival Aa injections), at the end of the 30-day experimental period. Each dot represents one sample. Oral microbiomes showed distinct clustering for each group (PERMANOVA, *p* < 0.05), except for SHAM and *Aa*+*Sg*. In the gut microbiomes, infected groups clustered apart from SHAM (*p* < 0.05), with *Aa*+*Sg* and *Sg* displaying differences. Ellipses in A show that SHAM and *Aa+Sg* oral microbiomes cluster apart from *Aa*- or *Sg*-infected groups. Ellipses in (**B**) show that SHAM gut microbiome clustered apart from the infected groups.

**Figure 6 ijms-25-08090-f006:**
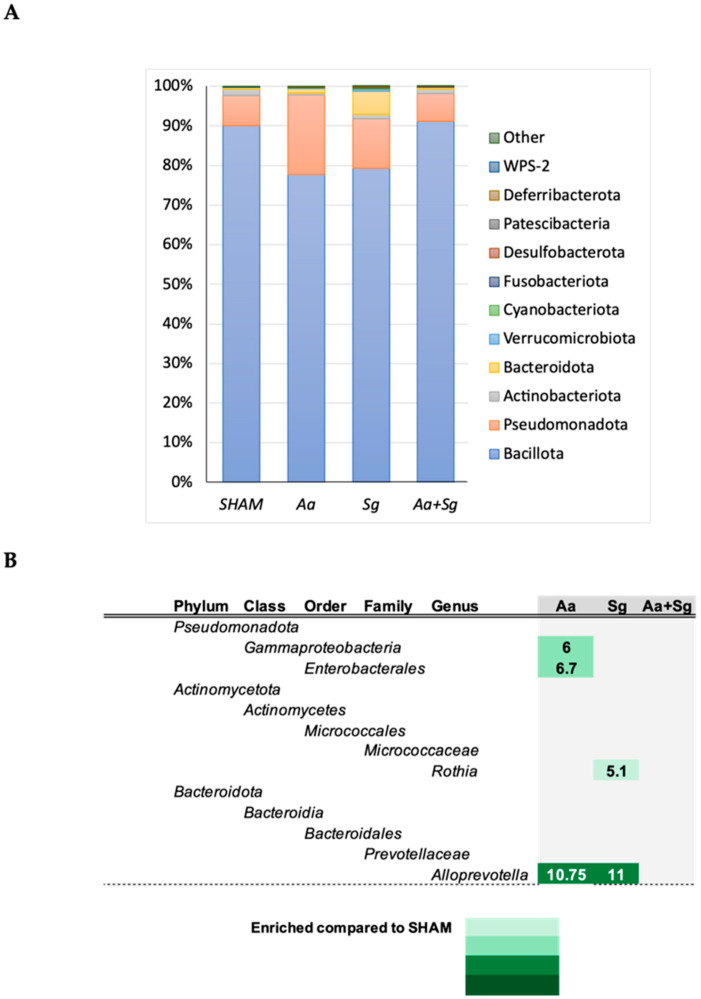
Oral microbiome of C57BL/6 mice subjected to various infection protocols showed differences in abundance of selected taxa. SHAM (negative control), *Aa* (oral and gingival *Aa* administration), *Sg* (oral *Sg* administration), and *Aa*+*Sg* (oral *Aa* and *Sg* administration with gingival Aa injections). Phyla relative abundance (%)-based plots (**A**). ANCOM revealed increased abundance of Gammaproteobacteria, Enterobacterales, *Alloprevotella*, and *Rothia* in each infected group compared to SHAM (**B**). The 75th percentile of the W distribution was used as the empirical cut-off value, and fold changes in abundances relative to the SHAM group are presented.

**Figure 7 ijms-25-08090-f007:**
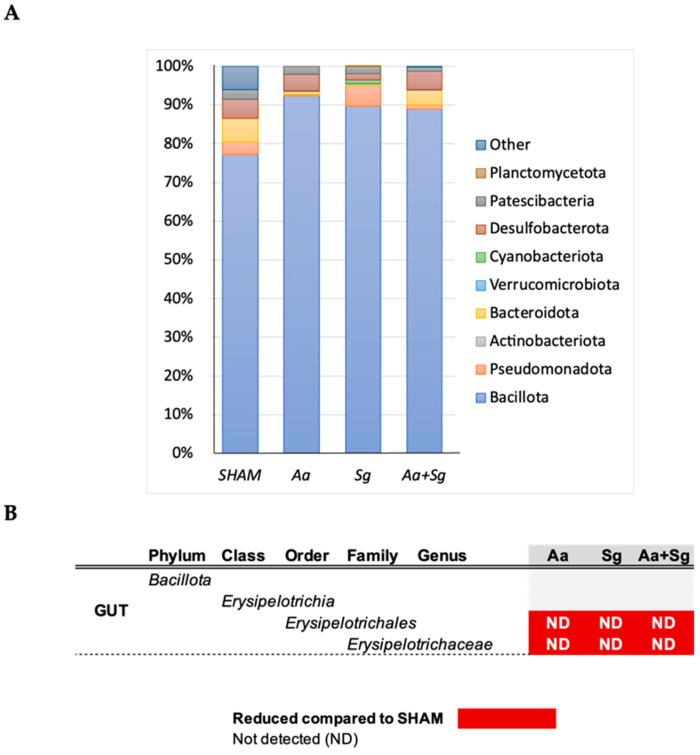
Gut microbiome of C57BL/6 mice subjected to various infection protocols showed differences in abundance of selected taxa. SHAM (negative control), *Aa* (oral and gingival *Aa* administration), *Sg* (oral *Sg* administration), and *Aa*+*Sg* (oral *Aa* and *Sg* administration with gingival Aa injections). Phyla relative abundance (%)-based plots (**A**). ANCOM revealed that the order Erysipelotrichales and family *Erisopelotrichaceae* were only detected in SHAM but not in the infected groups (**B**).

**Table 1 ijms-25-08090-t001:** Sequence of oligonucleotides used in Reverse Transcription qPCR for the determination of expression of TJ proteins and mucin-production-encoding genes.

Gene	Sequence (5′–3′)
*Cldn1* [93]	5′-AAGTGCTTGGAAGACGATGA-3′ 5′-AAGTGCTTGGAAGACGATGA-3′
*Cldn2* [94]	5′-ATACTACCCTTTAGCCCTGACCGAGA-3′ 5′-CAGTAGGAGCACACATAACAGCTACCAC-3′
*Ocdn* [93]	5′-CCAATGTGCAGGAGTGGG-3′ 5′-CGCTGCTGTAACGAGGCT-3′
*Muc1* [95]	5′-GTGCCCCCTAGCAGTACCG-3′ 5′-GACGTGCCCCCTACAATTGG-3′
*Muc2* [96]	5′-ACTGCACATTCTTCAGCTGC-3′ 5′-ATTCATGAGGACGGTCTTGG-3′
*Zo-1* [97]	5′-AGGACACCAAAGCATGTGAG-3′ 5′-GGCATTCCTGCTGGTTACA-3′

## Data Availability

Sequence data were submitted to Sequence Read Archive (SRA) under BioProject identification numbers PRJNA995344 and PRJNA994574.

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
