# Peer review of "Experimental Inoculation of *Aggregatibacter actinomycetemcomitans* and *Streptococcus gordonii* and Its Impact on Alveolar Bone Loss and Oral and Gut Microbiomes"

_ijms, 2024, doi:10.3390/ijms25158090_

Round 1
Reviewer 1 Report
Comments and Suggestions for Authors
The manuscript described the impact of oral inoculation of Aa, Sg and their association (Aa+Sg) on alveolar bone loss, oral microbiome and their potential effect on intestinal health in murine model.
The study designed and methods described found to be appropriate. The figures are self explanatory and very well documented. However I would like to make some suggestions
The title needs a correction. One suggestion is
Experimental inoculation of Aggregatibacter actinomycetemcomitans and Streptococcus gordonii and its impact on alveolar bone loss and oral and gut microbiomes;
Introduction: Line 62-64 'Aa's virulence is primarily...... The sentence is not clear. Please rephrase the sentence
Line 80.. typo 'thar'
Please provide the conclusive statement on the effect of inoculation of Aa and Sg on the alveolar bone loss.
Minor English revision is required through out the manuscript.
Comments on the Quality of English Language
Minor English revision is required through out the manuscript.
Author Response
To the Editor of IJMS
May 9th, 2024
Manuscript ID: ijms-2989982
“Experimental inoculation of Aggregatibacter actinomycetemcomitans and
Streptococcus gordonii on alveolar bone loss and oral and gut microbiomes.”
Dear Editor
We would like to express our thanks for the suggestions made by reviewers, which gave us an opportunity to improve the quality of the manuscript. We have modified the text according to the issues raised by the reviewers, as described below. To make it easier to see the changes in the main document they are marked in blue in the new version. Small grammar corrections were not marked to not pollute the document.
Our best regards
Luciana Saraiva and co-authors.
Reviewer 1.
Quality of English Language
() I am not qualified to assess the quality of English in this paper
( ) English very difficult to understand/incomprehensible
( ) Extensive editing of English language required
(x) Moderate editing of English language required
( ) Minor editing of English language required
( ) English language fine. No issues detected.
|
Yes |
Can be improved |
Must be improved |
Not applicable |
|
|
Does the introduction provide sufficient background and include all relevant references? |
(x) |
( ) |
( ) |
( ) |
|
Are all the cited references relevant to the research? |
(x) |
( ) |
( ) |
( ) |
|
Is the research design appropriate? |
(x) |
( ) |
( ) |
( ) |
|
Are the methods adequately described? |
(x) |
( ) |
( ) |
( ) |
|
Are the results clearly presented? |
( ) |
(x) |
( ) |
( ) |
|
Are the conclusions supported by the results? |
( ) |
(x) |
( ) |
( ) |
Comments and Suggestions for Authors
- The manuscript described the impact of oral inoculation of Aa, Sg and their association (Aa+Sg) on alveolar bone loss, oral microbiome, and their potential effect on intestinal health in murine model.
The study designed and methods described found to be appropriate. The figures are self-explanatory and very well documented. However, I would like to make some suggestions.
The title needs a correction. One suggestion is.
Experimental inoculation of Aggregatibacter actinomycetemcomitans and Streptococcus gordonii and its impact on alveolar bone loss and oral and gut microbiomes.
Answer: We thank the reviewer for his /her analyses of our manuscript. The title was changed as suggested.
- Introduction: Line 62-64 ‘Aa's virulence is primarily... The sentence is not clear. Please rephrase the sentence
Answer: The sentence was rephrased as follows:
“This organism is able to subvert host response due to the production of a leukotoxin and a cytolethal distending toxin (CDT)”. (with corrections, lines: 49-51)
- Line 80. typo 'thar'
Answer: changed to that. (with corrections, lines: 66)
Please provide the conclusive statement on the effect of inoculation of Aa and Sg on the alveolar bone loss.
Answer: a statement was added to the discussion. Lines 218 -221
“Thus, under the limitations of the murine model, it seems that Sg may be important for Aa survival in the supragingival biofilm, but the co-infection may not be relevant for Aa induced bone resorption.”
- Minor English revision is required throughout the manuscript.
Answer: the whole text was revised.

Reviewer 2 Report
Comments and Suggestions for Authors
A lot of information on the oral and gut microbiomes is included in the manuscript, which seems to be an intriguing read. There are a few minor comments that may be made before the manuscript is approved.
A more in-depth discussion of the importance of alveolar bone loss, as well as oral and gut microbiomes, should be included in the introduction.
.
Line 101: Reason for Aa and Aa+Sg showed reduced final weight compared to the Sg group.
Line 115: The author includes an explanation as to how they determined the amounts of LPS in the serum.
Place the figures in the correct alingent in Figure 3.
Remove lines no 130-165. It is recommended that the author use the IJMS format. The format was so jumbled up that it was impossible to read the paper.
Line 169: the author should in detail , Because two gut samples from the SHAM group did not produce amplicons, the author need to provide a detailed explanation for this failure.
Line 177: There are two lines that you should write regarding the significance of phylogenetic diversity.
The text size should be increased in figure 5, and the axis should be marked with a straight line.
Line 282: The author will explain how they will link the genes that are elevated and downregulated.
Author Response
To the Editor of IJMS
May 9th, 2024
Manuscript ID: ijms-2989982
“Experimental inoculation of Aggregatibacter actinomycetemcomitans and
Streptococcus gordonii on alveolar bone loss and oral and gut microbiomes.”
Dear Editor
We would like to express our thanks for the suggestions made by reviewers, which gave us an opportunity to improve the quality of the manuscript. We have modified the text according to the issues raised by the reviewers, as described below. To make it easier to see the changes in the main document they are marked in blue in the new version. Small grammar corrections were not marked to not pollute the document.
Our best regards
Luciana Saraiva and co-authors.
Reviewer 2.
Quality of English Language
() I am not qualified to assess the quality of English in this paper
( ) English very difficult to understand/incomprehensible
( ) Extensive editing of English language required
( ) Moderate editing of English language required
( ) Minor editing of English language required
(x) English language fine. No issues detected.
|
Yes |
Can be improved |
Must be improved |
Not applicable |
|
|
Does the introduction provide sufficient background and include all relevant references? |
( ) |
(x) |
( ) |
( ) |
|
Are all the cited references relevant to the research? |
(x) |
( ) |
( ) |
( ) |
|
Is the research design appropriate? |
( ) |
(x) |
( ) |
( ) |
|
Are the methods adequately described? |
(x) |
( ) |
( ) |
( ) |
|
Are the results clearly presented? |
( ) |
(x) |
( ) |
( ) |
|
Are the conclusions supported by the results? |
(x) |
( ) |
( ) |
( ) |
Comments and Suggestions for Authors
A lot of information on the oral and gut microbiomes is included in the manuscript, which seems to be an intriguing read. There are a few minor comments that may be made before the manuscript is approved.
A more in-depth discussion of the importance of alveolar bone loss, as well as oral and gut microbiomes, should be included in the introduction.
.
- Line 101: Reason for Aaand Aa+Sg showed reduced final weight compared to the Sg
Answer: We thank the reviewer for this intriguing question. Oral bacteria were related to inflammation and insulin resistance among diabetes‐free adults (Demmer et al. 2015). Furthermore, the detection of Aggregatibacter actinomycetemcomitans in the oral microbiome was associated with obesity in humans (Rahman et al., 2023).
The experimental inoculation of Aa in mice alters the gut microbiota and glucose metabolism (Komazaki et al, 2018). However, differing from our results, the experimental inoculation of Aa in mice given a high-fat diet (42% fat) (108 A. actinomycetemcomitans cells / 6 times per week for 6 weeks) led to increased weight gain but not in mice under normal chow diet (11%fat) when compared to their controls (Komazaki et al., 2018).
The reasons for the different results between Komazaki´s study and the present study may be dependent on the protocols. Mice lineage and sex have major effects on body weight gain in response to dietary and infection challenge (Milhem et al., 2021). The mice tested line did not differ (C57BL/mice), thus genetic background did not account for differences between our results and Komazaki et al data. We have used male mice, and Komazaki et al did not describe the sex of their tested mice. However, both studies differed on the inoculated Aa strain (ATCC 43718 versus JP2), the dose (1X108CFU versus 1X109 CFU), times per week (6 times versus 4 weeks) and via of inoculation of Aa (intragingival and oral versus oral inoculation). More importantly, Komazaki et al. 2018 administered norfloxacin (3 g/mouse) 6 times per week for 6 weeks to the mice, in order to “block the effect of A. actinomycetemcomitans”, whereas in the present study the mice resident microbiota was not altered by antimicrobials. It is already shown that norfloxacin treatment for two weeks can modulated the gut microbiota and leads to improved fasting glycemia and oral glucose tolerance (Chou et al., 2008). Thus, the results of both studies should not be compared.
As suggested by the reviewer, a short text was added to the discussion as follows (lines 174 to 182).
“The mice infected with Aa present lower weight at the end of the experimental period when compared with those infected with Sg or SHAM-infected mice. This observation differs from studies in humans where detection of Aa in the oral microbiome was associated with obesity (Rahman et al., 2023). Experimental administration of Aa in mice fed a high-fat diet also resulted in increased weight gain compared to non-infected mice, whereas infection had no effect in weight gain in animals under a normal chow diet (Komazaki et al, 2018). However, any comparison between our data and this latter study is prevented by different tested protocols, especially using norfloxacin, and antimicrobial that modulates the gut microbiota and leads to improved fasting glycemia and oral glucose tolerance (Chou et al., 2008). “
References:
Milhem A, Abu Toamih-Atamni HJ, Karkar L, Houri-Haddad Y, Iraqi FA. Studying host genetic background effects on multimorbidity of intestinal cancer development, type 2 diabetes, and obesity in response to oral bacterial infection and high-fat diet using the collaborative cross (CC) lines. Animal Model Exp Med. 2021 Feb 14;4(1):27-39. doi: 10.1002/ame2.12151.
Demmer RT, Jacobs DR Jr, Singh R, Zuk A, Rosenbaum M, Papapanou PN, Desvarieux M. Periodontal Bacteria and Prediabetes Prevalence in ORIGINS: The Oral Infections, Glucose Intolerance, and Insulin Resistance Study. J Dent Res. 2015 Sep;94(9 Suppl):201S-11S. doi: 10.1177/0022034515590369.
Chou CJ, Membrez M, Blancher F. Gut decontamination with norfloxacin and ampicillin enhances insulin sensitivity in mice. Nestle Nutr Workshop Ser Pediatr Program. 2008;62:127-37; discussion 137-40. doi: 10.1159/000146256.
Komazaki R, Katagiri S, Takahashi H, Maekawa S, Shiba T, Takeuchi Y, Kitajima Y, Ohtsu A, Udagawa S, Sasaki N, Watanabe K, Sato N, Miyasaka N, Eguchi Y, Anzai K, Izumi Y. Periodontal pathogenic bacteria, Aggregatibacter actinomycetemcomitans affect non-alcoholic fatty liver disease by altering gut microbiota and glucose metabolism. Sci Rep. 2017 Oct 24;7(1):13950. doi: 10.1038/s41598-017-14260-9. Erratum in: Sci Rep. 2018 Mar 12;8(1):4620. PMID: 29066788; PMCID: PMC5655179.
Rahman B, Al-Marzooq F, Saad H, Benzina D, Al Kawas S. Dysbiosis of the Subgingival Microbiome and Relation to Periodontal Disease in Association with Obesity and Overweight. Nutrients. 2023 Feb 6;15(4):826. doi: 10.3390/nu15040826. PMID: 36839184; PMCID: PMC9965236.
- Line 115: The author includes an explanation as to how they determined the amounts of LPS in the serum.
Answer: The method for evaluating Serum Lipopolysaccharide Levels was already described in Material and Methods, lines 408 to 414.
- Place the figures in the correct alignment in Figure 3.
Answer: done as suggested.
Remove lines no 130-165. It is recommended that the author use the IJMS format. The format was so jumbled up that it was impossible to read the paper.
Aswer: done as suggested.
- Line 169: the author should in detail, because two gut samples from the SHAM group did not produce amplicons, the author need to provide a detailed explanation for this failure.
Answer: As mentioned in the text, two gut samples from the SHAM group did not yield amplicons for analysis. DNA extraction was performed in six gut and six oral samples per group and sent for amplification. However, there were no amplicons in two of these gut samples. Clinical samples often pose limitations, such as low nucleic acid concentrations and the presence of inhibitors, thereby increasing the risk of sequencing failure. It is also possible that DNA was degraded during transportation.
- Line 177: There are two lines that you should write regarding the significance of phylogenetic diversity.
Answer: We added the following to the discussion section. (lines 277-384).
Our data indicate that oral administration of Sg increased the diversity of the oral microbiome (Faith Phylogenetic Diversity Index), whereas Aa or the consortium Aa+Sg reduced the diversity when compared to infection with Sg. Phylogenetic diversity accounts for the phylogenetic relatedness of the community members (Faith, 1992) and reduced diversity induced by Aa means that this organism put pressure on the oral microbiome in order to select a subset of organisms. These findings are consistent with studies showing a decrease in the microbial diversity of the oral cavity of periodontitis patients [82,83], indicating a role for Aa in oral dysbiosis.
- The text size should be increased in figure 5, and the axis should be marked with a straight line.
Answer: Done as suggested. Ellipses were also added to figure 5 to show clustering.
- Line 282: The author will explain how they will link the genes that are elevated and downregulated.
Answer:
The text in the discussion was changed as follows (Lines 251-264).
Oral infection with Aa and/or Sg resulted in altered transcription of several genes involved in gut barrier integrity [65]. Generally, Aa or Sg downregulated the expression of Claudin 1 and Claudin 2, Occludin, and Zonulin 1, whereas the Aa+Sg consortium upregulated the expression of Claudin 2, Occludin, and Zonulin 1. The elevated levels of Il-1β transcripts and low levels of il-6 transcripts in the gut of Aa-infected mice may partially account for this regulation induced by the pathogen [66]. Microbial insults and the resulting inflammatory response are known to contribute to the regulation of TJs expression and localization. Expression of Claudin 1 diminishes upon chronic exposure to LPS challenge [67], whilst Claudin-2 expression is upregulated by IL-22 and associated with protection of microbial-induced colitis [38]. IL-6 upregulates expression of claudin-2, while TNF, and IL-1β indirectly trigger occludin removal from the tight junction and increases leak pathway permeability. On the other hand, IL-10 and TGFβ promote epithelial proliferation and restitution, leading to barrier restoration [68]. Thus, the attenuation of Il-1β and Il-6 regulation observed when Aa was combined with Sg may contribute to the enhanced expression of TJs in the gut barrier. Previous data reported that Streptococcus gordonii exerts a beneficial effect on the epithelial barrier function, particularly in gingival cells, by inhibiting cytokine secretion and enhancing the expression of key tight junction components, thus reinforcing the oral barrier against microbial invasion [69]. However, it should be noticed that the changes in TJs transcription profiles induced by Aa, and /or Sg should be discreet, since we could not detect endotoxemia in any of the studied groups, an indicative of altered gut permeability.
References:
Fujita T, Firth JD, Kittaka M, Ekuni D, Kurihara H, Putnins EE. Loss of claudin-1 in lipopolysaccharide-treated periodontal epithelium. J Periodontal Res. 2012 Apr;47(2):222-7. doi: 10.1111/j.1600-0765.2011.01424.x.
Tsai PY, Zhang B, He WQ, Zha JM, Odenwald MA, Singh G, Tamura A, Shen L, Sailer A, Yeruva S, Kuo WT, Fu YX, Tsukita S, Turner JR. IL-22 Upregulates Epithelial Claudin-2 to Drive Diarrhea and Enteric Pathogen Clearance. Cell Host Microbe. 2017 Jun 14;21(6):671-681.e4. doi: 10.1016/j.chom.2017.05.009.
Abraham C, Abreu MT, Turner JR. Pattern Recognition Receptor Signaling and Cytokine Networks in Microbial Defenses and Regulation of Intestinal Barriers: Implications for Inflammatory Bowel Disease. Gastroenterology. 2022;162(6):1602-1616.e6. doi: 10.1053/j.gastro.2021.12.288.

Reviewer 3 Report
Comments and Suggestions for Authors
I would not recommend the publication of this research article. I do not find the theme of the article interesting and as per the standards of the reputed journal. The abstract is very poorly written. The introduction section has not presented the theme of the research article with sufficient citations. The experimentation section is very weak and the obtained results are not sufficient enough for publication. Graph quality is also not of good resolution. Discussion is very inefficient. Altogether, I would not recommend the publication of this research article.
Comments on the Quality of English LanguageExtensive english editing is required.
Author Response
Manuscript ID: ijms-2989982
“Experimental inoculation of Aggregatibacter actinomycetemcomitans and
Streptococcus gordonii on alveolar bone loss and oral and gut microbiomes.”
Dear Editor
We would like to express our thanks for the suggestions made by reviewers, which gave us an opportunity to improve the quality of the manuscript. We have modified the text according to the issues raised by the reviewers, as described below. To make it easier to see the changes in the main document they are marked in blue in the new version. Small grammar corrections were not marked to not pollute the document.
Our best regards
Luciana Saraiva and co-authors.
Reviewer 3.
Quality of English Language
() I am not qualified to assess the quality of English in this paper
( ) English very difficult to understand/incomprehensible
(x) Extensive editing of English language required
( ) Moderate editing of English language required
( ) Minor editing of English language required
( ) English language fine. No issues detected.
|
Yes |
Can be improved |
Must be improved |
Not applicable |
|
|
Does the introduction provide sufficient background and include all relevant references? |
( ) |
( ) |
(x) |
( ) |
|
Are all the cited references relevant to the research? |
( ) |
( ) |
(x) |
( ) |
|
Is the research design appropriate? |
( ) |
( ) |
(x) |
( ) |
|
Are the methods adequately described? |
( ) |
( ) |
(x) |
( ) |
|
Are the results clearly presented? |
( ) |
( ) |
(x) |
( ) |
|
Are the conclusions supported by the results? |
( ) |
( ) |
(x) |
( ) |
Comments and Suggestions for Authors
I would not recommend the publication of this research article. I do not find the theme of the article interesting and as per the standards of the reputed journal. The abstract is very poorly written. The introduction section has not presented the theme of the research article with sufficient citations. The experimentation section is very weak, and the obtained results are not sufficient enough for publication. Graph quality is also not of good resolution. Discussion is very inefficient. Altogether, I would not recommend the publication of this research article.
Comments on the Quality of English Language
Extensive English editing is required.
Answer: The whole text was carefully revised. The figures were improved, and all sections were changed.

Round 2
Reviewer 3 Report
Comments and Suggestions for Authors
I would recommend the publication of this article